# The Complex Role of HBeAg and Its Precursors in the Pathway to Hepatocellular Carcinoma

**DOI:** 10.3390/v15040857

**Published:** 2023-03-27

**Authors:** Kiyasha Padarath, Aurélie Deroubaix, Anna Kramvis

**Affiliations:** 1Hepatitis Virus Diversity Research Unit, Department of Internal Medicine, School of Clinical Medicine, Faculty of Health Sciences, University of the Witwatersrand, 7 York Road, Parktown, Johannesburg 2193, South Africa; 2Life Sciences Imaging Facility, Faculty of Health Sciences, University of the Witwatersrand, 7 York Road, Parktown, Johannesburg 2193, South Africa

**Keywords:** hepatitis B virus, HBeAg, hepatocellular carcinoma

## Abstract

Hepatitis B virus (HBV) is one of the seven known human oncogenic viruses and has adapted to coexist with a single host for prolonged periods, requiring continuous manipulation of immunity and cell fate decisions. The persistence of HBV infection is associated with the pathogenesis of hepatocellular carcinoma, and various HBV proteins have been implicated in promoting this persistence. The precursor of hepatitis e antigen (HBeAg), is translated from the precore/core region and is post-translationally modified to yield HBeAg, which is secreted in the serum. HBeAg is a non-particulate protein of HBV and can act as both a tolerogen and an immunogen. HBeAg can protect hepatocytes from apoptosis by interfering with host signalling pathways and acting as a decoy to the immune response. By evading the immune response and interfering with apoptosis, HBeAg has the potential to contribute to the hepatocarcinogenic potential of HBV. In particular, this review summarises the various signalling pathways through which HBeAg and its precursors can promote hepatocarcinogenesis via the various hallmarks of cancer.

## 1. Introduction

Hepatitis B is caused by the hepatitis B virus (HBV), a hepatrotropic DNA virus belonging to the family *Hepadnaviridae* [1]. Despite the availability of an effective vaccine, hepatitis B continues to be a major health concern, with over 296 million people chronically infected with HBV in 2019 and 1.5 million new infections each year, accounting for 820,000 deaths annually as a result of severe liver disease, including cirrhosis and hepatocellular carcinoma (HCC; liver cancer) [2].

HBV infection is linked to more than 60% of all HCC in developing countries and 40% in developed countries, thus implicating HBV infection as a carcinogen [3]. Carcinogenesis is a multistep process in which normal cells are transformed into cancer cells by acquiring various properties that allow them to form tumours [4]. The acquired properties of cancer cells that distinguish them from normal cells have been classified as a series of hallmarks of cancer [5]. These hallmarks of cancer include: sustaining proliferative signalling, evading growth suppressors, resisting cell death, enabling replicative immortality, inducing angiogenesis, deregulation of cellular energetics, genome instability and mutation, tumour-promoting inflammation, avoiding immune destruction, activating invasion, and metastasis (Figure 1) [6]. HBV infection can promote liver cancer through direct or indirect mechanisms. HBV infection can:generate genome instability by integrating its viral DNA into the host genome [7,8]cause abnormal expression of oncogenes and tumour suppressor genes leading to activation of cancer-related pathways, interference with epigenetic and chromosomal remodelling through integration [7,8].affect the liver microenvironment through interference with both innate and adaptive immunity and thus help the virus evade immune surveillance, increasing persistence and disease progression to tumour formation [7,9].interfere with mitochondrial proteins, to change mitochondrial dynamics/signalling, leading to the development of HCC [10,11].persist in liver tissue, increasing chronic oxidative damage in hepatocytes, immune-mediated inflammation of the liver, and development of HCC [12].interfere with apoptosis signalling to promote viral proliferation and HCC progression [8,13].

The 3.2 kb HBV genome has four overlapping reading frames and encodes for the five major RNA transcripts [14,15,16,17], which are responsible for the translation of seven viral proteins: HBcAg (capsid), polymerase (pol), envelope proteins, which have three different sizes—small (SHBs), medium (MHBs), and large (LHBs)—and non-particulate proteins HBx and HBeAg [18,19]. HBx and HBsAg have been implicated in the hepatocarcinogenic potential of HBV. The HBx protein has been shown to be involved in the transcriptional activation of cellular growth regulatory genes, modulation of apoptosis, angiogenesis, and metastasis, inhibition of nucleotide excision repair of damaged cellular DNA, interaction with cellular proteins, and activation of cell signalling pathways [20,21,22,23]. Accumulation of preS1 large envelope proteins and/or preS2/S mutant proteins in the endoplasmic reticulum (ER) increases ER stress, which has been shown to activate the unfolded protein response and ultimately the development of HCC [24]. Individuals infected with preS mutant HBV have a 13X risk of developing HCC [25].

HBeAg is a non-structural, secreted protein expressed by every member of the family *Hepadnaviridae*, though its expression is not required to maintain infection. The antigen has been used clinically as an index of viral replication, infectivity, the severity of disease, and response to treatment [9,26,27]. The presence of HBeAg can delineate the different phases of chronic hepatitis B virus infection [28]. Although the majority of the protein is secreted, approximately 20–30% is retained in the cytoplasm [9,26]. HBeAg has the dual role of being a tolerogen and an immunogen [9]. The exact mechanisms by which HBeAg achieves these opposing functions is not fully understood. Its immune-modulatory role allows for humoral and cell-mediated immune evasion of HBV [26,27]. In addition, studies have shown that HBeAg can protect hepatocytes from apoptosis, leading to the survival of infected hepatocytes and contributing towards the development of HBV chronicity [29,30,31]. 

By evading the immune response and interfering with apoptosis, HBeAg has the potential to contribute to the hepatocarcinogenic potential of HBV. In this review, we discuss the biogenesis of HBeAg, its interaction with immune cells, and the different signalling pathways. We propose ways in which this protein and its precursors can promote hepatocarcinogenesis through the various hallmarks of cancer (Figure 1). 

## 2. Biogenesis of HBeAg

The core protein, HBcAg, is the structural component of the viral capsids, found in mature and incomplete virions. HBeAg, a non-structural HBV protein, is the N-terminal processed product of the precore/core protein, with different forms of HBeAg precursors. The 25 kDa precursor of HBeAg (p25) is translated from the preCore/Core open reading frame (ORF) on the precore mRNA, whilst HBcAg is translated from the Core ORF on the pregenomic RNA (pgRNA) (Figure 2) [32,33,34]. The translation of the HBeAg precursor p25 from precore mRNA starts at the first ATG site, while the translation of the core protein on pgRNA starts at the second ATG site [7,35,36]. p25 is translated and targeted to the ER membrane for post-translational modifications [37,38,39]. In the ER, the 29 amino acid N terminal signal of p25 is proteolytically removed to create the precursor HBeAg protein p22 [40,41]. Due to the presence of a nuclear localisation signal (NLS) on the arginine-rich C-terminal domain (CTD) of p22, this protein can also be transported into the nucleus [39,42]. In vitro studies using subgenomic p22 confirmed the protein’s nuclear and cytoplasmic localisation, but the role of p22 in the nucleus remains unknown [43]. The p22 protein undergoes further modification in the ER through the cleavage of the CTD by a furin endoprotease to give rise to the mature HBeAg, which is also known as p17 [44,45,46,47]. The mature HBeAg is secreted in a soluble form in the blood serum [48,49].

HBeAg can form part of a collection of three species of proteins known as HB core related antigen (HBcrAg), which consists of related proteins sharing an identical 149 amino acid sequence: HBcAg, HBeAg, and a truncated 22 kDa precore protein (p22Cr). Like HBeAg, p22cr is also a processed product of the precore protein, but with protein processing only occurring at the C-terminal and not the amino end [50] (Figure 2). Recent studies have shown that the detection of HBcrAg levels is a useful marker for disease monitoring, predicting treatment response and disease outcome in chronic hepatitis B (CHB), and can also predict the risk of HCC [28,51,52].

## 3. Effect of BCP/Precore Mutations on the Expression of HBeAg

There are multiple mutations in the basal core promoter (BCP) or precore region that affect the expression of HBeAg at transcriptional, translational, and post-translational levels [53]. The tendency of the HBV genome to develop these mutations is dependent on the HBV (sub)genotypes, as the sequence of the preC/C region differs in the different (sub)genotypes [54,55]. For example, 1858C in the preCore/Core(PreC/C) region is positively associated with genotypes A, F, and H and 1858T with genotypes B, D, and E [55]. Subgenotype C2 has 1858T as opposed to C1 that has 1858C, and F1/F4 can be differentiated from F2/F3 by having 1858T instead of 1858C, while 1888A is positively associated with subgenotype A1 [55]. It has been suggested that the different mutations in preC/C can account for the different clinical manifestations and duration of HBV infection in individuals infected with the different (sub)genotypes [53,54].

Genotypes with T at position 1858 are frequently associated with the 1896A mutation, which converts the codon for tryptophan to a stop codon. This premature stop codon leads to the truncation of the HBeAg precursor and abrogation of HBeAg expression, which is responsible for HBeAg to anti-HBeAg seroconversion as a result of the truncation of the precore/core fusion protein [56]. The G1896A mutation has been reported in patients with chronic liver disease and severe liver damage in Africa and Asia [57,58]. The G1896A mutation occurs in up to 30% of all HBV sequences and is frequent in (sub)genotypes C1, D, E, and F [59,60].

The BCP double mutation 1762T/1764A can affect transcription of precore mRNA and result in decreased levels of HBeAg [61,62], without switching off HBeAg expression. This mutation occurs in approximately 25% of all HBV sequences [63] and is significantly more frequent in genotypes A and C compared to D and B [64,65]. A high prevalence of 1762T/1764A has been detected in HBV strains from HCC patients compared to asymptomatic carriers of HBV [61]. Genotype D develops 1762T/1764A and the precore stop codon mutation G1896A [66,67]. Patients infected with genotype D seroconvert from HBeAg to anti-HBe in adolescence or early adulthood, leading to high chronicity [68,69], which can lead to cirrhosis and HCC [70].

Subgenotype A1 has unique BCP/precore mutations that affect the expression of HBeAg at a transcriptional, translational, and post-translational level [32,71]. The first is the presence of the 1762T/1764A mutation; this mutation has been shown to reduce HBeAg expression in an additive manner and is associated with a higher risk of cancer [32,61,71]. The second mutation is the 1809–1812 TCAT mutation in the Kozak region preceding the precore ORF, which converts the region from an optimal to a suboptimal translation context. This mutation causes decreased translation of HBeAg by a ribosomal leaky scanning mechanism [72,73]. Lastly, in the precore region there is a G to T transversion at position 1862. This mutation results in a valine to phenylalanine substitution. Phenylalanine interferes with the signal peptide cleavage, resulting in decreased expression and secretion of HBeAg, as well as the accumulation of its precursor (p22) in the cytoplasm of the hepatocyte [29,74,75]. Thus, in subgenotype A1, HBeAg loss and immune evasion are the result of a number of mutations that HBV has evolved to survive, persist, and infect [53,76]. Furthermore, subgenotype A1 has been associated with severe liver disease and a 4.5-fold increased risk of developing HCC at an earlier age compared to other HBV genotypes [32,53]. 

The BCP/precore mutations can be found in 94.4% of HBV isolates from HCC cases; therefore, it is extremely important to understand and investigate the downstream effects of these mutations on hepatocarcinogenesis [77]. The expression of HBeAg is affected at two levels by these mutations: reduction or abrogation of HBeAg expression, removing the tolerogenic effect of HBeAg and change of expression and localisation of HBeAg and/or its precursors in the cellular compartments, which can promote endoplasmic reticulum (ER) stress and/or apoptosis. These changes can promote carcinogenesis [78]. 

## 4. HBeAg-Associated Hallmarks of Cancer

The HBeAg-associated hallmarks of cancer, depicted in blue in Figure 1, will be discussed.

### 4.1. Immune Evasion, Leading to Persistence

The dual roles of HBeAg as a tolerogen and immunogen and its ability to suppress or activate the immune response reveal the complexity of the interactions between HBeAg and the host [9]. Numerous studies have shown that HBeAg can impact both the innate and adaptive immune responses, in order to promote HBV persistence.

Macrophages, which exhibit phenotypic plasticity, form the first line of defence against HBV. They can release either pro-inflammatory cytokines (M1 polarisation), or anti-inflammatory cytokines (M2 polarisation) [79]. Kupffer cells, found in the sinusoids of the liver, form the largest macrophage population in the human body and play an important role in liver inflammation. Previous exposure to HBeAg can affect the response of Kupffer cells to HBV and play a key role in the induction of macrophage activation, which may exacerbate liver injury by increasing the production of inflammatory cytokines [80,81]. Macrophages from mice that have previously been exposed to maternal HBeAg, undergo M2 polarisation, with increased programmed Death Ligand 1 (PD-L1) expression in both Kupffer and cytotoxic T-cells. Increased PD-L1 expression plays an important role in T cell exhaustion and impairment of the cytotoxic T lymphocyte (CTL) response in chronic HBV (CHB) patients [82]. In the absence of maternal HBeAg in the mice, Kupffer cells undergo M1 polarisation and can inhibit viral replication by the activation of CD8+ T cells [83]. These experiments demonstrate the importance of HBeAg as a tolerogen because both M2 macrophage polarisation and increased PD1 expression lead to HBV persistence [83]. Lower levels of baseline soluble PD1 (sPD1) are associated with HBeAg seroclearance following two years of antiviral treatment in HBeAg-positive CHB patients [84] (Figure 3).

HBeAg can activate the mitogen-activated protein kinase (MAPK) pathway, including activation of extracellular signal-regulated kinase (ERK), c-Jun N-terminal kinase (JNK), and p38 in macrophages, and influence the production of interleukin-6 (IL-6) and tumour necrosis factor alpha (TNF-α) [85]. Overexpression of miR-212-3p, located on chromosome 17q13.3, which has been shown to exert tumour-suppressive roles by targeting MAPK pathways in prostate cancer [86], leads to decreased mRNA and protein expression of pro-inflammatory cytokines IL-6 and TNF-α [85,86]. In HBeAg-stimulated human and mouse Kupffer cells, miR-212-3p is also upregulated [85]. However, this process is not uncontrolled, and HBeAg-induced macrophages can regulate the expression of miR-212-3p via the MAPK pathway; specifically, ERK acts as an inhibitor and blocks the expression of miR-212-3p, whereas p38 and JNK increase its expression [85]. In addition, cAMP response element-binding protein (CREB), an important transcription factor, is activated in HBeAg-induced macrophages and inhibits the expression of miR-212-3p via the ERK/CREB pathways [85]. Thus, although HBeAg promotes the expression of miR-212-3p in macrophages, miR-212-3p expression can be regulated via the ERK/CREB signalling pathways by negative modulation of inflammatory cytokine production (Figure 3) [85]. 

Alternatively, HBeAg can regulate cytokine production in macrophages by modulating nuclear factor kappa-light-chain-enhancer of activated B cells (NF-κB) and P13K signalling pathways. In macrophages, HBeAg induces expression of miR-155, located on chromosome 21q21.3 and associated with inflammation and immunity [87]. miR-155 is induced by the activation of Toll-like receptor (TLR) ligands, in particular TLR4 and TLR3 [88,89]. The increased miR-155 represses the expression of SH-2 containing inositol 5’ polyphosphatase 1 (SHIP-1), leading to *Akt* phosphorylation upon lipopolysaccharide (LPS) stimulation [90] and inflammatory response to LPS through targeting SOCS-1 [91]. Thus, HBeAg-induced miR-155 functions as a positive regulator in HBeAg-induced expression of inflammatory cytokines by targeting BCL-6, SHIP-1, and SOCS-1 in macrophages [87]. miR-155 also directly represses the expression of BCL-6, a transcription factor that downregulates NF-κB signalling in mouse macrophages [87,91,92]. Thus, by inducing miR-155, HBeAg can downregulate interleukin1 (IL-1) and reduce the activation of NF-κB [87]. In fact, increased NF-κB levels were found in liver biopsies of HBeAg-negative patients compared to HBeAg-positive patients, and HBeAg-activation of macrophages via the TLR-2/NF-κB signal pathways, where HBeAg binds to the TLR2, has been found in patients with exacerbated hepatic fibrosis [87,93]. HBeAg has also been shown to directly downregulate NK cell-mediated gamma interferon (IFN-γ) production and interleukin-18 (IL-18) signalling [94] (Figure 3). 

In addition, studies have shown that HBeAg could suppress both Toll-like receptor 2 (TLR2) and IL-1β-induced activation of NF-κB in cells and clinical samples. A study by Wang and colleagues identified that HBeAg can associate with NF-κB essential modulator (NEMO) [95]. NEMO is a scaffolding component of the IκB kinase complex required for NF-κB activation in vitro [96]. HBeAg can associate with NEMO, and interfere with the activation of NF-κB, thus reducing inflammation and increasing virus replication. Viral mutations such as G1862T and G1896A, which reduce or abrogate HBeAg expression, respectively, can lead to the removal of the tolerogenic effect of HBeAg, leading to the activation of the immune response [97]. As a result, loss of HBeAg in patients infected with HBeAg negative mutants can induce higher levels of NF-κB activity compared to HBeAg positive patients [95]. 

HBeAg can also bind to peripheral blood monocytes (PBMC), neutrophils, and B lymphocytes but not to T lymphocytes [98]. The interaction between HBeAg and monocytes and neutrophils was shown to be dose-dependent, leading to the inhibition of both cell types [98]. Monocytic myeloid-derived suppressor cells (mMDSCs) are derived from myeloid progenitor cells and comprise of only ~0.5% of PBMCs in healthy individuals. The mMDSC population expands during infection, inflammation, and cancer [99]. HBeAg plays an important role in mMDSC population expansion and induction of immune tolerance [100]. Compared to HBeAg negative patients, HBeAg-positive patients were shown to have significantly higher levels of mMDSCs [100]. When PBMCs from healthy patients were exposed to HBeAg, there was an increase in mMDSCs and the expression of IL-6 and IL-1β. In addition, mMDSCs from HBeAg-positive patients reduced the proliferation of CD4+ and CD8+ T cells [100]. This may be a possible mechanism by which HBeAg modulates the host immune response during CHB by physically depleting or weakening virus-specific CD4 and CD8 T cells. These cells are thus unable to proliferate in response to the viral antigens or produce important antiviral and immunostimulatory cytokines, which are critical for the control of the virus in CHB patients [101] (Figure 3). 

The TLR family is part of the pattern recognition receptor (PRR) family and is responsible for recognising diverse molecules derived from pathogens and damaged host cells, also called pathogen-associated molecular patterns (PAMPs) [102]. HBeAg down-regulates the expression of the TLR family within Kupffer cells and blood monocytes [103,104]. When peripheral blood monocyte cells were pre-treated with HBeAg, they displayed an impaired TLR signalling response [104]. TLR-2 has been shown to play an important role in the recognition of HBV and, once activated, induces the expression of INF and proinflammatory cytokines by macrophages [104]. HBeAg positive patients have lower levels of TLR-2 than HBeAg negative patients [104]. Patients who lost HBeAg after interferon therapy had increased TLR-2 levels and increased expression of IL-6 [105]. In addition, the HBeAg precursor, p22, has been shown to bind to toll/IL-receptor (TIR) and disrupt TLR signalling [106]. These results indicate that HBeAg has a negative role in the regulation of TRL2, which may be beneficial in HBV replication and persistence. Furthermore, HBeAg-containing conditioned medium but not hepatitis B core antigen (HBcAg)-containing conditioned medium suppresses the p38 MAPK phosphorylation induced by TLR2- or TLR4- agonists in blood CD14+ monocytes [104]. In another study, the cytosolic-form HBeAg suppressed TLR-mediated NF-κB activation by targeting the Toll/IL-receptor containing proteins TRAM and Mal [106]. These findings imply that HBeAg may have a potential to reduce TLR signalling in monocytes [98] and promote viral persistence. 

Although the precise mechanism is unknown, HBeAg may regulate T cell-mediated immune responses. Circulating HBeAg preferentially activates Th2 cells and has the potential to deplete HBeAg- and HBcAg-specific Th1 cells, which are required for viral clearance. This depletion can lead to the persistence of HBV [29,30,107]. 

#### JAK-STAT Pathway

Alpha interferon (IFN-α) is used clinically to treat CHB patients; however, only approximately one-third of patients respond to IFN-α treatment [108]. Patients with low HBeAg levels prior to treatment are more likely to respond to IFN-α therapy, and HBeAg negative patients with core promoter mutations respond better to IFN-α therapy than HBeAg positive patients [109,110], suggesting that HBeAg or its intracellular precursors may interfere with the IFN signalling representing a viral strategy to persist in the host through induction of immune tolerance. 

Two mechanisms have been proposed to explain the reason for the reduced response to IFN treatment in the presence of HBeAg (Figure 3).

Mitra and colleagues hypothesised that the reduced antiviral activity of IFN-α against HBV could be due to blockade of the IFN-elicited JAK-STAT pathway by a HBV protein [111]. In their study HBeAg-positive patients exhibited weaker induction of interferon stimulated genes (ISGs) in their livers than HBeAg-negative patients upon IFN-α therapy. Furthermore, the cytosolic HBeAg precursor p22 protein significantly reduced interferon-stimulated response element (ISRE) activity and the expression of ISGs upon IFN-α stimulation in cell culture [111]. p22 did not alter the total STAT1 or pSTAT1 levels in cells treated with IFN-α, but instead blocked the nuclear translocation of pSTAT1 by interacting with karyopherin α1 (Kα1) through its CTD domain, thus impeding JAK-STAT signalling, resulting in host innate immune response evasion and causing resistance to IFN therapy [111].Yu and colleagues hypothesised that HBeAg reduced IFN effectiveness and enhanced HBV infection by hijacking the IFN/JAK/STAT pathway [112]. Members of the intracellular suppressor of cytokine signalling (SOCS) family are regulators of cytokine signalling pathways [113,114] that are induced by cytokines and act in a classic negative-feedback loop to inhibit cytokine signalling [115]. SOCS1 and SOCS3 inhibit interferon-mediated antiviral and antiproliferative activities [116,117]. In their study, Yu and colleagues revealed that HBeAg initially activates SOCS2 through the ERK pathway. HBeAg-activated SOCS2 subsequently reduces tyrosine kinase 2 (TYK2) stability, down-regulates IFN receptor expression, represses STAT1 phosphorylation, and finally attenuates ISGs production [112]. Thus, revealing a novel mechanism by which HBeAg is coordinated to enhance HBV replication by hijacking the IFN/JAK/STAT pathway and antiviral action.

HBeAg interaction with the innate and adaptive immunity is multifaceted and complex. The main role of HBeAg is to direct the immune system away from HBV hepatocytes to ensure immune evasion, which is a hallmark of cancer. HBeAg can lead to the downregulation of pro-inflammatory transcription factors such as NF-κB, JAK/TYK/STAT, and P13K. These activated transcription factors mediate the expression of key cytokines and chemokines as well as inflammatory enzymes during HBV infection, leading to persistence. 

**Figure 3 viruses-15-00857-f003:**
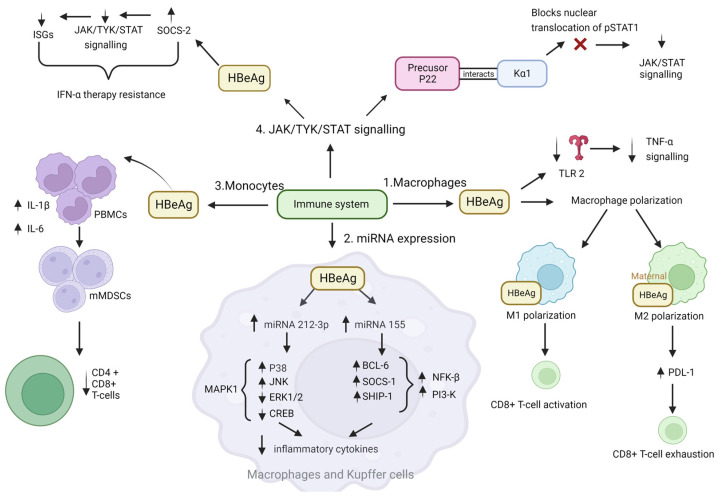
The role of HBeAg in immune evasion: influence of HBeAg on different components of the immune system that HBeAg can influence: macrophages, monocytes, miRNA expression, and JAK/TYK/STAT signalling. 1. Macrophages—exposure of macrophages to HBeAg can influence the type of macrophage polarisation Kupffer cells undergo [84]. In addition, HBeAg can influence macrophages by decreasing TLR 2 [104]. 2. miRNA expression—HBeAg leads to an increase in miRNA-212-3p and miRNA-155, which leads to a decrease in inflammatory cytokines and immune evasion [85,87]. 3. Exposure of PBMCs monocytes to HBeAg resulted in an increase in mMDSCs, the expression of IL-6 and IL-1β, and reduced proliferation of CD4+ and CD8+ T cells [100]. 4. JAK/TYK/STAT pathway—HBeAg can influence the JAK/TYK/STAT pathway via two mechanisms to decrease response to INF therapy. In the first mechanism, the HBeAg precursor p22 blocked the nuclear translocation of pSTAT1 by interacting with Kα1, thus impeding JAK/TYK/STAT signalling [111]. In the second mechanism, mature HBeAg activates, which represses STAT1 phosphorylation, thus impeding JAK/TYK/STAT signalling [112]. Image created using biorender.com.

### 4.2. Tumour Promoting Inflammation

Hanahan and Weinberg included tumour-promoting inflammation as one of the enabling characteristics of cancer [5]. There are several types of inflammation that can promote cancer development and progression, differing by cause, mechanism, outcome, and intensity [118]. Approximately 20% of human cancers may be related to chronic inflammation caused by infections, exposure to irritants, or autoimmune disease [119]. HBV infection can persist in liver tissue, increasing immune-mediated inflammation of the liver and the development of HCC [120]. The main role of HBeAg is tolerogenic, to promote persistence and decrease the expression of inflammatory cytokines. During HBV infection, viral diversity increases during the course of infection, with the emergence of strains defective in HBeAg expression [78]. Various BCP/precore mutations can affect the expression of HBeAg, resulting in decreased HBeAg expression, mislocalisation of HBeAg and its precursors, or complete loss of HBeAg. As a result, during HBV infection, there is a progression from the HBeAg positive phase, which is associated with high virus replication and low inflammation, to the HBeAg negative phase, which is associated with chronic infection and more inflammation in the liver [78,121]. Defective expression of HBeAg results in the immune system and inflammatory cytokines being directed towards HBV infected hepatocytes, which may explain why BCP/precore mutations are more frequently associated with HCC [53,78,122].

### 4.3. Resisting Cell Death

Apoptosis is the cell’s natural mechanism for programmed cell death [123] and plays a critical role in development as well as homeostasis [124]. There are two different pathways that lead to apoptosis: the intrinsic and extrinsic pathways. Intrinsic apoptosis occurs when an injury, such as direct DNA damage, hypoxia, or deprivation of growth factors, occurs within the cell. Whilst extrinsic apoptosis occurs, the cell uses extracellular signals, such as cytotoxic stress and receptor-mediated pathways, to induce apoptosis [125].

#### 4.3.1. Intrinsic Mechanisms

p53 driven apoptosis

p53 is a critical activator of the intrinsic apoptotic pathway. p53, a tumour suppressor located in the nucleus, plays a critical role in determining whether the DNA will be repaired or if the damaged cell will undergo apoptosis. MDM2 (the murine double minute 2) is a protein that regulates p53 function by degrading it and inhibiting its location in the nucleus [126]. Regions in MDM2 and p53 are sensitive to exposure to environmental carcinogens and can lead to the development of HCC. Mutations in the MDM2-p53 axis and chronic HBV infection have been shown to trigger the development of HCC [127]. A study by Liu and colleagues identified a previously unidentified mechanism in which HBeAg and its precursor p22 may play an important role in tumour growth and hepatocellular carcinogenesis [128]. This study showed that NUMB Endocytic Adaptor Protein (NUMB), a cell fate determinant, antagonises the activity of the plasma membrane receptor of the NOTCH family [129]. NUMB positively regulates p53 by forming a tricomplex with p53 and the E3 ubiquitin ligase HDM2 (also known as MDM2) and thus preventing the ubiquitination and degradation of p53 [129]. One recognised function for NUMB is its involvement in inhibition of the NOTCH pathway, which plays a role in carcinogenesis. NUMB inhibits Notch signalling through interaction with the Notch intracellular domain and promotion of its ubiquitination and degradation [130]. NUMB was also shown to control the intracellular trafficking of the Notch to suppress its function. Studies also indicated that the expression of NUMB correlates with the suppression of hedgehog signalling (Shh) by downregulating the activated Notch1 protein and thereby inhibiting the mRNA expression of Shh [131]. The NUMB protein was shown to interact with the precursor p22 and mature HBeAg at the 150–165 amino acid regions (CTD), with no interaction observed between HBcAg and NUMB [128]. HBeAg/precore could associate with NUMB and impair the stability and transcriptional activity of p53, thereby enhancing carcinogenesis. This carcinogenic effect was as a result of HBeAg’s antiapoptotic effect caused by compromised p53 activity rather than deregulated NOTCH activity [128]. Besides compromising p53 activity, it has been shown that HBeAg/precore can inhibit NF-κB activity and expression of its downstream inflammatory cytokines and chemokines, whereas HBV variants causing reduced HBeAg expression lead to a higher level of inflammation compared to the wild-type [106]. Analysis of the clinical samples from HCC patients with the G1896A mutation had lower p53 activity than those with the wild-type [128]. HBeAg-deficient HBV variants may cause a greater inflammatory response compared with the wild type, and can worsen liver diseases with increased possibility of developing HCC. This is a possible explanation for why HBeAg-negative mutations can also have a higher risk of HCC [26,132] (Figure 4).

p25 regulation of apoptosis via UPR

HBV precore variants, such as the G1862T mutation, result in HBeAg accumulation in the cytoplasm of Huh7 cells. During post-translational modifications of the p25 precursor of HBeAg, the G1862T mutation prevents signal peptide cleavage of the amino end, causing accumulation in the ER leading to the prevention of synthesis and secretion of HBeAg [29,133]. Furthermore, the retro-transport of the misfolded G1862T preC/C precursor protein from the ER to the cytosol for degradation results in increased expression of ubiquitin and proteasomes and the formation of ubiquitin-rich mini-aggregates along the microtubules in the cytosol [29,74,134]. 

The ER quality control (QC) mechanism can thus be compromised when unfolded or misfolded viral proteins accumulate and aggregate in the lumen of the ER, resulting in ER stress [135,136]. In order to overcome this ER stress, the cell activates the unfolded protein response (UPR) or ER stress response [136,137,138]. Once initiated, it results in activation of signalling cascades in a sequential order by three ER-localised transmembrane transducers, which include: (1) double-stranded RNA-dependent protein kinase (PKR)-like ER kinase (PERK), (2) activating transcription factor (ATF) 6, and (3) inositol-requiring enzyme (IRE1) [137,138]. 

These signalling cascades alter the cell’s transcriptional and translational programmes to cope with stressful conditions and resolve protein-folding defects, thus ensuring cell survival. If the UPR fails, cells initiate lethal programmes such as autophagy or cell death by either apoptosis or necrosis [136,139,140]. In a study by Bhoola and colleagues, the activity of the PERK and IRE1/XBP1 pathways appeared to be decreased, whereas the ATF6 pathway appeared to be increased, while apoptosis was unchanged for cells transfected with the G1862T mutant when compared to the wild-type counterpart. Therefore, HBeAg in patients infected with the G1862T mutant leads to increased ER stress, which can result in liver damage, which has been shown to be a contributing factor to HCC [122]. A higher frequency of G1862T has been found in both serum and liver tissue of HCC patients compared to asymptomatic carriers [141] (Figure 4). 

#### 4.3.2. Extrinsic Pathways

HBeAg regulation of apoptosis via Fas/FasL

Hepatic apoptosis during HBV infection is mostly mediated by death signalling from members of the tumour necrosis factor (TNF) protein family, including TNF-α, Fas ligand (FasL), and TNF-related apoptosis-inducing ligand (TRAIL) [142]. Fas, a glycoprotein, is present in the liver cell membrane and triggers apoptosis signals to the hepatocytes when Fas ligand (FasL) binds with it [143]. FasL can induce massive hepatic apoptosis in most types of liver disease and is increased in patients with chronic and acute hepatitis B or HBV-related hepatocellular carcinoma [144,145]. The Fas/FasL apoptosis signal is strictly regulated at both receptor and intracellular apoptotic signalling events. Previous studies suggest that Fas transcription is activated by p53 in hepatocytes, and the cross talk between the Fas signalling and p53 in controlling apoptosis is of clinical importance [146,147]. 

As shown previously, HBeAg and its precursors may function as a significant enhancer of HCC through many mechanisms, including repressing p53 activity through interacting with NUMB [128,148]. A study by Liu and colleagues showed that HBeAg serves to inhibit p53-dependent FasL- and tumour necrosis factor-related apoptosis-inducing ligand (TRAIL)-mediated hepatic apoptosis by down-regulating both membrane (m) Fas death receptors DR4 and DR5 expression at a transcriptional level and enhancing the expression of soluble (s) Fas by increasing Fas alternative splicing [148]. The results indicate that HBeAg can protect hepatocytes from apoptosis induced by the Fas/FasL and TRAIL/DR systems [148]. Therefore, HBeAg likely possesses negative immunomodulatory roles [30,31]. These findings show that inhibition of the Fas and TRAIL apoptotic signals by HBeAg can lead to the survival of infected hepatocytes and contribute towards the development of HBV chronicity and hepatocarcinogenesis (Figure 4).

The loss of apoptotic control is a hallmark of cancer because it allows cancer cells to survive longer and allows for the accumulation of mutations, which can increase invasiveness during tumour progression, stimulate angiogenesis, deregulate cell proliferation, and interfere with differentiation [149]. Although through different mechanisms, the role of HBeAg in these apoptotic pathways helps promote the survival of hepatocytes infected with HBV.

**Figure 4 viruses-15-00857-f004:**
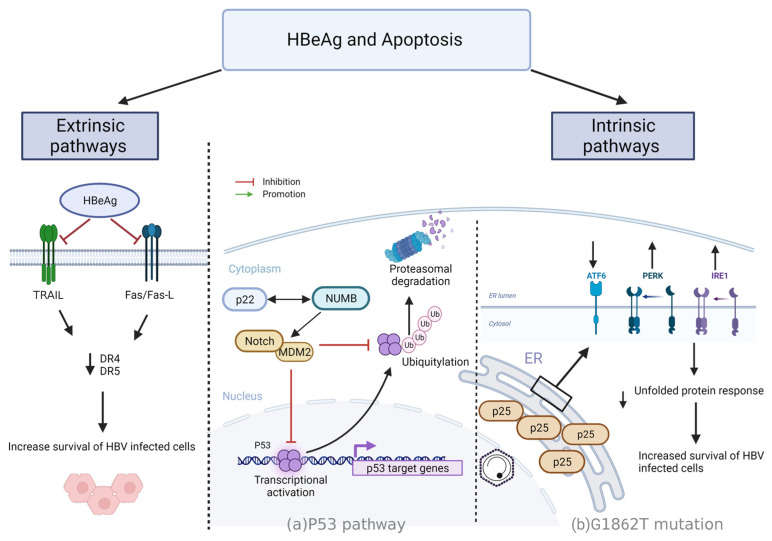
The role of HBeAg in Apoptosis: the role of HBeAg and its precursors in extrinsic and intrinsic apoptotic pathways. Extrinsic pathway: HBeAg inhibits Fas/FasL and TRAIL/DR systems which leads to the down-regulation of Fas death receptors, DR4 and DR5, expression to promote the survival of HBV infected hepatocytes [148]. Intrinsic pathways: (**a**) P53 pathway—p22 interacts with NUMB, which regulates Notch and MDM2 preventing ubiquitination and degradation of p53 and impairing the stability and transcriptional activity of p53 [128]; (**b**) G1862T mutation—G1862T mutation results in the accumulation of p25 in the ER, decreasing the activity of the PERK and IRE1/XBP1 while increasing the activity of the ATF6 receptor pathways. This results in a decrease in UPR to promote survival of HBV infected hepatocytes [122]. Image created using biorender.com.

### 4.4. Promote Sustained Proliferative Signalling

Cancer cells can proliferate indefinitely. Cells have a receptor at their surface which, when receiving a growth factor signal from neighbouring cells, initiates an intracellular cascade of signalling resulting in cell growth and division [150]. Cells normally need this feedback from other cells to know when to divide, ensuring that proliferation happens in a coordinated manner.

#### 4.4.1. HBeAg Interferes with Wnt/β-catenin Signalling

The *Wnt* signalling pathway had been identified and studied extensively in carcinogenesis and is regulated via multiple mechanisms [151]. The most common occurrence has been investigated in colon cancer, where it results in constitutively active *Wnt* signalling downregulation by naturally occurring inhibitors. This leads to increased or decreased transcription of *T-cell factor (TCF)/β-catenin* target genes downstream [152,153]. 

Liver cancers, including HCC, display similar *Wnt*-related mechanisms to colon and gastric cancer [153]. Studies have shown that constitutively active *Wnt* signalling in HCC is primarily via mutations to the *CTNNB1* gene that remove the regulatory phosphorylation sites from the N-terminus of β-catenin [154]. A recent study by Tran and colleagues screened various HBV proteins for their impact on *Wnt* signalling. The study showed that HBeAg precursor p22 had a strong correlation with *Wnt* signalling. The p22 protein was more potent than the HBx protein, which is a known and well-studied carcinogen. Other pre-core/core proteins (p25, p21, or p17) did not activate TCF/β-catenin transcription despite significant amino acid sequence overlap with p22 [155].

The study also demonstrated that Frizzled-7 (Fzd7) and Glutamate-Ammonia Ligase (GLUL), which have a known association with liver cancer and interact with TCF/β-catenin, are induced by p22 in vivo [155]. Furthermore, they demonstrated that p22 can increase *TCF/β-catenin* transcription on its own and in conjunction with ectopically expressed wild-type or mutant β-catenin. Activation of *TCF/β-catenin* transcription by p22 was blocked by dnTCF4, confirming its impact specifically on *Wnt* signalling [155] (Figure 5).

Selective regulation of *Wnt* signalling is central to identifying possible therapeutic interventions for cancer. Further investigations need to be done to determine if hyperactive *Wnt* signalling is due to loss of transcriptional repression or activation of transcription [155]. 

#### 4.4.2. HBeAg Interferes with the Regulation of the Cell Cycle

Increasing evidence is emerging that the expression of cellular miRNA and their gene targets play a prominent role in HBV pathogenesis [21,156]. miRNA are small non-coding RNAs of 19–23 nucleotides that play key roles in the regulation of almost every cellular process in all multicellular eukaryotes [157]. Oncogenic miRNAs may facilitate cell cycle entry and progression by targeting CDK inhibitors or transcriptional repressors of the retinoblastoma family. On the other hand, tumour suppressor miRNAs induce cell cycle arrest by downregulating multiple components of the cell cycle machinery [158]. Viruses can be affected by these miRNAs, whilst other viruses, such as herpesvirus, encode their own miRNA to increase replication and evade the immune system [159]. HBV can modulate cellular miRNA to achieve the same results [21]. 

HBx, is an oncoprotein that is a major regulator of both gene expression and miRNA expression in the host [21], and it has been well documented to interact with various miRNA. HBeAg could also influence miRNA expression in a manner similar to HBx. Indeed, clinical studies have indicated differences in miRNA profiles between HBeAg positive and negative patients. miRNAs identified with disrupted plasma expression specifically in HBeAg-positive patients with chronic hepatitis and with liver-specific target genes are biomarkers for disease progression and might impact the development of HCC [160]. This has paved the way for extending research into the association of HBeAg and miRNA regulation. 

Cell-cycle progression is stimulated by protein kinase complexes, each of which consists of a cyclin and a cyclin-dependent kinase (CDK), to ensure faithful replication of the genome before cell division [161]. A study by Samal and colleagues showed that HBeAg is an oncoprotein that regulates the cell cycle through the induction of miR-106b. MicroRNA-106b is a member of the miR-106b-25 cluster and is known to act as an oncogene that is associated with poor prognosis among HCC patients [94]. Samal and colleagues used anti-miR-106b (anti-sense oligonucleotides that inhibit miR-106b) to show an increase in retinoblastoma gene (Rb) gene expression in Huh7 cells, suggesting that the *Rb* gene is a target of miR-106b at 3′UTR (3′ untranslated region) of the *Rb* gene. The *Rb* gene is a tumour suppressor gene with a well-documented role in HCC, including the modulation of cell proliferation and cell cycle progression [94]. Anti-miR-106b inhibits cell proliferation in HBeAg expressing Huh7 cells but not in control Huh7 cells and these findings clearly highlight the link between HBeAg-induced miR-106b [162]. By targeting the *Rb* genes, this leads to enhanced cell proliferation and promotes progression from the G0/G1 phase into the S phase of the cell cycle in Huh7 cells [162]. This suggests that HBeAg is an oncoprotein that inhibits the *Rb* gene expression via induction of an oncogenic miRNA, miR-106b (Figure 5).

**Figure 5 viruses-15-00857-f005:**
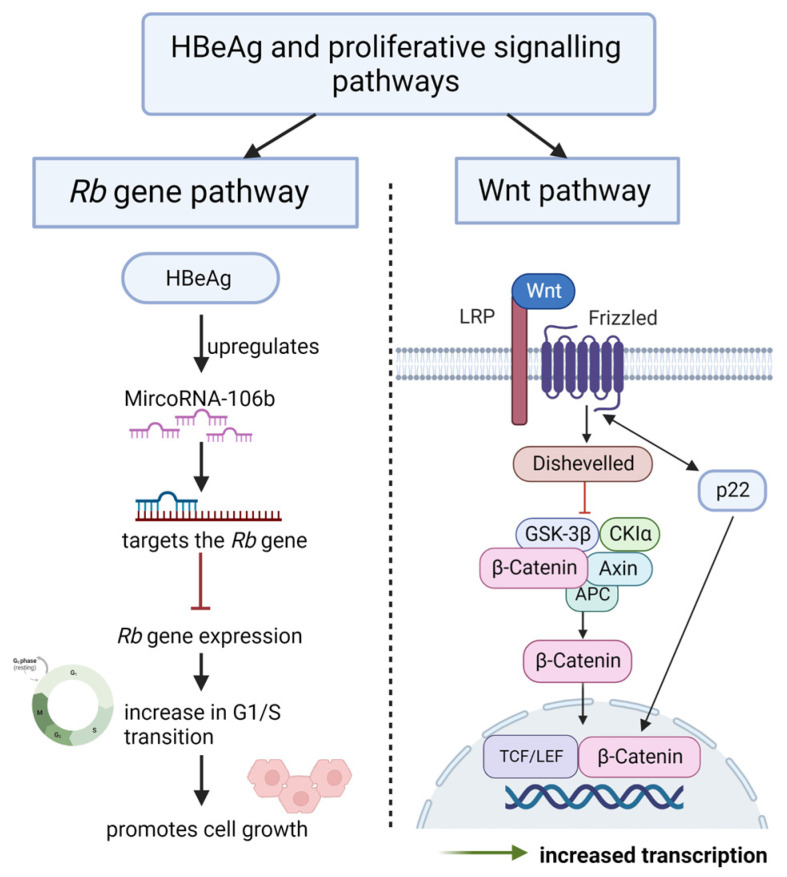
Role of HBeAg in proliferative signalling pathways: the role of the HBeAg *Rb* gene pathway and the wnt signalling pathway. *Rb* gene pathway—HBeAg upregulates miR-106b, which targets the Rb gene. Thus, reducing *Rb* gene expression, resulting in a promotion of progression from the G0/G1 phase into the S phase of the cell cycle in Huh7 cells, adapted with permission from [162], Scientific reports (http://creativecommons.org/licenses/by/4.0/, accessed on 23 March 2023). Wnt signalling pathway—p22 induces Fzd7 and GLUL and is able to increase TCF/β-catenin transcription on its own to promote proliferative signalling [155]. Image created using biorender.com.

## 5. Discussion

HBV is one of the seven known human oncogenic viruses and has adapted to coexist with a single host for prolonged periods, requiring continuous manipulation of immunity and cell fate decisions [163,164]. Oncogenic viruses promote tumorigenesis using relatively few multifunctional viral oncoproteins [165,166].

HBx protein is the most well-researched HBV oncoprotein. As shown for human papillomavirus (HPV) and simian vacuolating virus 40 (SV40), a virus can have several viral oncoproteins [166]. Thus, the complex interlinking roles of HBeAg and its precursors, which we have reviewed, may implicate HBeAg and its precursors as potential HBV oncoproteins. However, this would need to be confirmed by studies in in vivo animal models, as has been done for HBx and HBsAg [20,21,167].

In summary, mutations of the BCP/precore region that result in loss of extracellular HBeAg and accumulation of various HBeAg precursors in the cytoplasm of HBV infected cells [29,53,78]. HBeAg and its precursors can lead to the downregulation of pro-inflammatory transcription factors such as NF-κB, JAK/TYK/STAT, and P13K [132,164]. These activated transcription factors mediate the expression of key cytokines and chemokines to direct the immune system away from HBV-infected hepatocytes and ensure immune evasion [111]. Secondly, defective expression of HBeAg results in the immune system and inflammatory cytokines being directed towards HBV-infected hepatocytes, which cause an increase in inflammation [74,78]. Thirdly, HBeAg and its precursors interfere with both intrinsic and extrinsic apoptotic pathways to ensure the survival of HBV-infected hepatocytes [122,128,148]. Lastly, HBeAg and its precursors, can initiate an intracellular signalling cascade, which leads to cell growth and division [155,162]. Therefore, this review has demonstrated that the HBeAg and its precursors can play roles in at least four hallmarks of cancer: avoiding immune destruction; tumour-promoting inflammation; resisting cell death; and sustained proliferative signalling (Figure 1). Thus, these roles of HBeAg and its precursors implicate them as potential oncoproteins.

Further research on the role of HBeAg and its precursors in carcinogenesis is ongoing. These studies could discover mechanisms by which HBeAg and its precursors can be associated with additional hallmarks of cancer. For example, it has been shown that p22 and p25 can translocate to the nucleus [43], where they could possibly interact with the host genome, leading to genomic instability and mutation and possibly enabling the replication of immortal liver cells. Fifty years after its discovery [49], HBeAg continues to intrigue, with its unique properties, and keep researchers active in elucidating its intricate role in the molecular biology and pathogenesis of HBV infection, in particular its role in hepatocarcinogenesis.

## Figures and Tables

**Figure 1 viruses-15-00857-f001:**
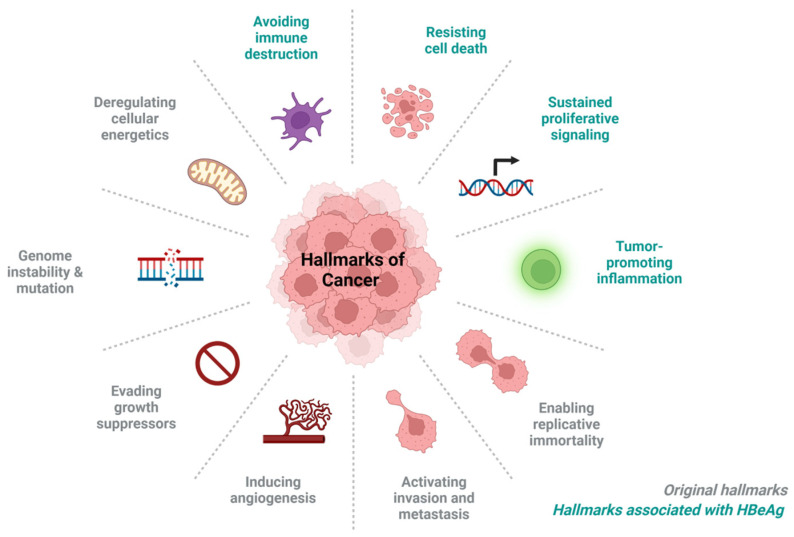
Hallmarks of cancer: HBeAg associated hallmarks of cancer are indicated in blue. Image created using biorender.com.

**Figure 2 viruses-15-00857-f002:**
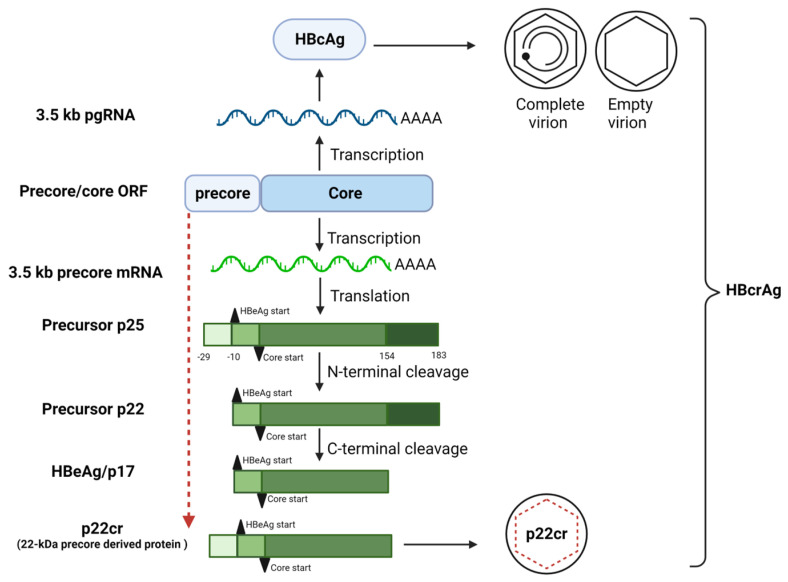
Biogenesis of HBeAg and its precursors. HBeAg and HBcAg translations from the preCore/Core ORF, adapted with permission from [50], © 2021 American Society for Microbiology. Image created using biorender.com.

## Data Availability

Not applicable.

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
