# Peer review of "The Complex Role of HBeAg and Its Precursors in the Pathway to Hepatocellular Carcinoma"

_viruses, 2023, doi:10.3390/v15040857_

Round 1

Reviewer 1 Report

There are several HBV proteins that have been shown to promote HCC via direct or indirect mechanisms. In this review, Padarath et al. summarized the role of HBeAg in hepatocarcinogenesis and proposed HBeAg and its precursors as potential HBV oncoproteins. The manuscript is comprehensive, well organized and uses adequate references. One question regarding clinical relevance of this manuscript: how does the authors explain that HBV patients with BCP/precore mutations with low or lack of HBeAg expression lead to a high chronicity and liver cancer? This can be interpreted that not HBeAg but intracellular HBeAg precursors or HBcAg can promote HCC.

Author Response

Thank you for taking the time to review our work. Please see the attachment 

Reviewer 2 Report

Authors review the literature surrounding HBeAg’s role in liver oncogenesis and organize it according to the Hallmarks of Cancer framework. Overall the manuscript reads well. I have several suggestions listed in attached file, but most are minor.

Author Response

(The authors gave the same response as above.)

Reviewer 3 Report

In this review, Kiyasha and colleagues summarized the multi-faceted roles of HBeAg and its precursors in the development of HCC by incorporating the hallmarks-of-cancer framework originally developed by Hanahan and Weinberg. This review is quite comprehensive as it covers most of the published concepts on the biogenesis of HBeAg, clinical variations that leads to its aberrant expression or function, immune-regulatory functions and molecular interactions that are relevant to cancer formation.

At the end of the introduction section, the authors wrote: “In this review, we discuss the biogenesis of HBeAg, its interaction with immune cells and the different signalling pathways, and we propose ways in which this protein and its precursors can promote hepato-carcinogenesis through the various hallmarks of cancer (Figure 1).” However, the title of this manuscript seems to deviate from the main content. There are objective criteria for defining a substance as a carcinogen. In terms of viral proteins, the most convincing data would be the phenotypes of the transgenic mice expressing the protein of interest. In this regard, the HBx and large S protein of HBV are well-qualified as cancer promoting agents. HBeAg or precore protein on the other hand lack such well-supported proof as a hepatocarcinogen. Indeed, the authors stated at the end of section 4.2 that “Defective expression of HBeAg results in the immune system and inflammatory cytokines being directed towards HBV infected hepatocytes, and could possibly be the cause of why BCP/precore mutations are more frequently associated with HCC”. This showed that it is not the expression of HBeAg but rather the deactivation or aberrant expression of precore protein that promotes the development of HCC. In summary, I think the Question, i.e., “Are HBeAg and its precursors potential hepatocarcinogens?” is a pseudo-proposition and the main text does not intend to prove or disprove it. Up till now, there is generally very limited evidence supporting HBeAg or its derivatives as hepatocarcinogen.  Rather, its expression, mutation, or genetic inactivation perturb the viral and host activity throughout the long process of CHB-induced liver cancer. In this sense, HBeAg and its precursors may modify the chronicity of HBV infection and played a complex role in the pathway to HCC.

Apart from the main concerns, there are some other minor issues:

1. In the last sentence of the abstract, “this review identifies the various signalling pathways…”. I believe these signalling pathways were identified by the large number of researchers cited in this review but not by the author of this manuscript. Hence, it is prudent to use “summarize” or an equivalent.

2.There are issues with reference, the majority of the text follows the numbered reference style while another style was used in line 390. In addition, many of the listed references lack journal name.

3. There is an error in the author’s name, in line 366, it should be “Liu” as opposed to “Lui”.

Author Response

(The authors gave the same response as above.)
